# Cumulative effect of PM$_{2.5}$ components is larger than the effect of PM$_{2.5}$ mass on child health in India

Ekta Chaudhary [1,10], Franciosalgeo George[2,10], Aswathi Saji[2], Sagnik Dey [1,3,4] ✉, Santu Ghosh [5] ✉, Tinku Thomas [5], Anura. V. Kurpad [6], Sumit Sharma[7], Nimish Singh [1,7], Shivang Agarwal[7,8] & Unnati Mehta[9]

While studies on ambient fine particulate matter (PM$_{2.5}$) exposure effect on child health are available, the differential effects, if any, of exposure to PM$_{2.5}$ species are unexplored in lower and middle-income countries. Using multiple logistic regression, we showed that for every 10 μg m$^{-3}$ increase in PM$_{2.5}$ exposure, anaemia, acute respiratory infection, and low birth weight prevalence increase by 10% (95% uncertainty interval, UI: 9–11), 11% (8–13), and 5% (4–6), respectively, among children in India. NO$_3^-$, elemental carbon, and NH$_4^+$ were more associated with the three health outcomes than other PM$_{2.5}$ species. We found that the total PM$_{2.5}$ mass as a surrogate marker for air pollution exposure could substantially underestimate the true composite impact of different components of PM$_{2.5}$. Our findings provide key indigenous evidence to prioritize control strategies for reducing exposure to more toxic species for greater child health benefits in India.

Air pollution poses a significant global health risk, with 6.67 million (95% UI: 5.90–7.49) deaths worldwide attributable to the combined effects of household and ambient air pollution[1]. In 2019, over 99% of the world's population inhaled fine particulate matter (PM$_{2.5}$) concentrations that exceeded the World Health Organization (WHO) annual air quality guideline (AQG) of 5 μg m$^{-3}$ [2]. In India, home to one-sixth of the global population, ambient PM$_{2.5}$ exposure has been rising in the last three decades[3]. In 2017, the majority (76.8%) of the people in India were exposed to an annual population-weighted mean PM$_{2.5}$ higher than the national ambient air quality standard (NAAQS) of 40 μg m$^{-3}$ [4].

The latest round of the state-level burden of diseases in India estimated that ambient PM$_{2.5}$ exposure was responsible for 0.98 million (0.77–1.19) deaths and 17.8 million disability-adjusted life years[5]. The *Lancet* Commission has emphasized that children under the age of 5 years (U5) are more vulnerable to the harmful impacts of air pollution,

even at low levels, particularly during foetal development and the first few months of life[6]. Thus, early-life exposure to air pollution[7] impacts child health outcomes such as acute respiratory infection (ARI)[8], which are the most common cause of global childhood morbidity and mortality[9]. In addition to mortality, ambient PM$_{2.5}$ exposure is a risk factor for adverse pregnancy outcomes, including low birth weight (LBW)[10], stillbirth[11], foetal mortality[12], preterm birth[13], and birth defects[14]. In early childhood, it is a risk factor for impaired child growth, stunting[6,15], and anaemia[16,17]. Since the lower and middle-income countries (LMICs) have higher levels of ambient PM$_{2.5}$ exposure compared to higher-income countries; the LMIC children are likely to have disproportionately higher rates of these morbidities and mortality linked to prenatal and early childhood exposure to ambient PM$_{2.5}$[18].

PM$_{2.5}$, by itself, is a composite mixture of multiple chemical species. The components of PM$_{2.5}$ may have varying toxicity with varying mass fractions. As a result, the true cumulative impact of ambient PM$_{2.5}$

[1]Centre for Atmospheric Sciences, Indian Institute of Technology Delhi, New Delhi, India. [2]Division of Epidemiology, Biostatistics, and Population Health, St John's Research Institute, Bangalore, India. [3]Centre of Excellence for Research on Clean Air, IIT Delhi, New Delhi, India. [4]School of Public Policy, IIT Delhi, New Delhi, India. [5]Department of Biostatistics, St John's Medical College, Bengaluru, India. [6]Department of Physiology, St John's Medical College, Bengaluru, India. [7]TERI, New Delhi, India. [8]Johns Hopkins University, Baltimore, MD, USA. [9]Harvard T.H. Chan School of Public Health, Boston, USA. [10]These authors contributed equally: Ekta Chaudhary, Franciosalgeo George. ✉e-mail: sagnik@cas.iitd.ac.in; santu.g@stjohns.in

on health could be manifold higher than that estimated by the total $PM_{2.5}$ mass[19,20]. Moreover, the $PM_{2.5}$ species can be tagged to specific emitting sectors, providing opportunities to examine their relative importance in mitigation strategies. The association between $PM_{2.5}$ components and their sectoral contributions and U5 child health outcomes is unknown at the national level in the LMICs.

To address these critical knowledge gaps, we estimated the association of ambient $PM_{2.5}$ and its components and emitting sectors on three U5 child health outcomes—anaemia, ARI, and LBW, in India and provided a hypothesis to establish a causal inference of the results. We combined anthropometric measurements, blood biomarkers, and socioeconomic information from the fourth round of the national family health survey (NFHS-4) with granular information on sector-specific speciated $PM_{2.5}$ exposure that was obtained by integrating satellite-derived $PM_{2.5}$ with outputs from the weather research fore-casting (WRF) and community multi-scale air quality modelling system (WRF-CMAQ) model (see 'Methods' for details). In this study, we assess the effect of $PM_{2.5}$ components and their contributing sources on multiple child health outcomes and further assess the expected health benefits of meeting various clean air targets.

## Results

### Study population and characteristics

There were 259,627 observations in the original NFHS-4 dataset, of which 15,119 children had missing age. After removing missing records of exposure, outcome, and covariates, there were 177,072 observations in the final analytical sample for LBW and ARI (a detailed flow-chart is given in Supplementary Fig. 1). In the anaemia analysis, all the anaemia status-missing observations were excluded.

The national prevalence of LBW from the analytical sample was 16.6% (16.4, 16.8). This was slightly higher in girls (17.8%) than in boys (15.5%). A significant variation in the prevalence of LBW was observed across the levels of the mother's education, socioeconomic status, mother's body mass index (BMI), age, place of residence, and different levels of $PM_{2.5}$. The prevalence of anaemia among U5 children was 56.8% (56.6, 57.1). Anaemia was associated with maternal education, religion, socioeconomic, place of residence, levels of $PM_{2.5}$ exposure, and maternal anaemia status.

The national prevalence of ARI was 2.8% (2.7–2.9). Maternal education, religion, socioeconomic, place of residence, and levels of $PM_{2.5}$ exposure were significantly associated with ARI. The estimated ARI

prevalence was relatively higher in rural areas than in urban (2.9% vs. 2.4%) areas (Supplementary Table 1).

The annual average $PM_{2.5}$ level at the PSU level was $62\,\mu g\,m^{-3}$ with an interquartile range (IQR) of $52–79\,\mu g\,m^{-3}$. The most dominant $PM_{2.5}$ components were organic carbon (OC), $NO_3^-$, $NH_4^+$, $SO_4^{2-}$, and others that include chloride, sodium, magnesium, potassium, calcium, soil, and water molecules, and the remaining unspecified components. These were mostly contributed from domestic, industrial, interna-tional, agricultural, and transport sectors (Supplementary Table 2).

### Effects of $PM_{2.5}$ and its components and sources on child health

The two-stage model ('Methods') estimated the odds ratio, OR of LBW as 1.15 (1.12–1.18), of anaemia as 1.57 (1.54–1.59), and of ARI as 1.32 (1.24–1.4) for every IQR increase in ambient $PM_{2.5}$ exposure. Further, gestational exposure to $NO_3^-$ showed the highest association on LBW of children with OR 1.17 (1.14–1.2) per IQR increase in exposure. The estimated ORs of LBW for every IQR increase in gestational exposure of others, $NH_4^+$, elemental carbon (EC), soil, $SO_4^{2-}$, and OC were 1.14 (1.11–1.17), 1.13 (1.11–1.16), 1.11 (1.08–1.14), 1.09 (1.07–1.11), 1.07 (1.04–1.09), and 1.05 (1.03–1.08), respectively (Fig. 1). For anaemia, we observed the largest impact of $NO_3^-$ (OR: 1.36, 1.32–1.41), followed by $NH_4^+$ (OR: 1.28, 1.25–1.31), others (OR: 1.25, 1.21–1.28), EC (OR: 1.21, 1.18–1.25), soil (OR: 1.18, 1.16–1.20), $SO_4^{2-}$ (OR: 1.14, 1.12–1.17), OC (OR: 1.12, 1.09–1.15) (Fig. 1). Similarly, the effect of $NO_3^-$ on ARI was relatively higher than other components. The estimated OR of ARI was 1.52 (1.42–1.61) for every IQR increase in $NO_3^-$. EC also had a larger impact on ARI (OR: 1.49, 1.4–1.58), followed by OC (OR: 1.46, 1.37–1.55), others (OR: 1.33, 1.26–1.41) and $NH_4^+$ (OR: 1.15, 1.09–1.21) (Fig. 1).

Among the eight $PM_{2.5}$ sources analysed using the two-stage model, we found that the IQR increase in $PM_{2.5}$ from road dust exhibited a higher effect on LBW (OR: 1.13, 1.11–1.14), followed by the international transboundary transport (OR: 1.09, 1.07–1.1), the industry sector (OR: 1.07, 1.05–1.08), the agricultural sector (OR: 1.06, 1.05–1.07), other sectors (OR: 1.04, 1.02–1.07) and transport sector (OR: 1.05, 1.02–1.07). For anaemia, every IQR increase in $PM_{2.5}$ expo-sure attributable to the sectoral emissions of unorganized (that includes municipal waste burning and crematorium) sectors (OR, 1.19; 1.18–1.20) showed the highest adverse effect, followed by the inter-national transboundary transport (OR: 1.11, 1.09–1.13), domestic and road dust (ORs: 1.09 (1.06–1.11) and 1.09 (1.08–1.11), respectively), agriculture (OR: 1.08, 1.07–1.09), industry (OR: 1.04, 1.02–1.06), and

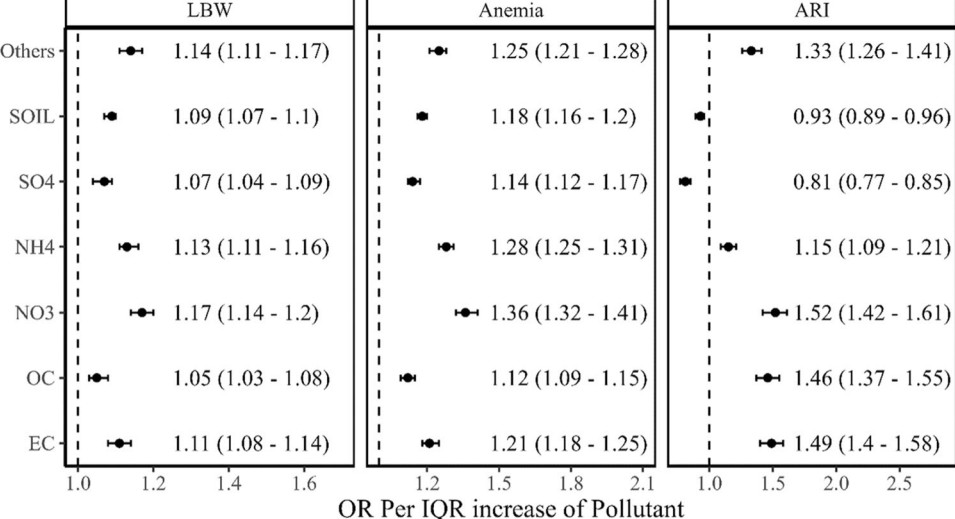

**Fig. 1 | Association between exposure to $PM_{2.5}$ components and child health outcomes.** Odds ratio per IQR of $PM_{2.5}$ components concentrations with their 95% confidence intervals (black dots with error bars) for given health outcomes i.e., LBW, Anaemia, and ARI. The analytical sample size used for each health outcome is provided in the consort diagram (Supplementary Fig. 1).

transport (OR: 1.03, 1.0–1.06) sector. The power sector (OR: 0.96, 0.94–0.98) did not show any effect on anaemia among children (Fig. 2). The domestic sector showed the largest effect on ARI than the rest of the sectors. For every IQR increase of $PM_{2.5}$ from the domestic sector, we observed OR of ARI as 1.30 (1.24–1.35) followed by the transport sector (OR: 1.21, 1.14–1.28), other sectors (OR: 1.21, 1.14–1.28) and the agricultural sector (OR: 1.1, 1.07–1.13).

### Effects of $PM_{2.5}$ components on anaemia and ARI stratified by LBW

Stratified estimates were obtained by adding an interaction term of LBW in the second-stage model with air pollutants, one at a time ('Methods'). For an IQR increase in $NO_3^-$, we observed the OR of anaemia as 1.47 (1.41–1.52) for children born with LBW vs. 1.34 (1.3–1.38) for children born with normal weight. Apart from $NO_3^-$, a larger difference was observed for EC (OR 1.3 vs. 1.2), and OC (OR 1.2 vs.

1.1). Similarly, OR of ARI was 1.72 (1.6–1.85) and 1.46 (1.37–1.56), respectively, for LBW and normal birth weight children for IQR increase in $NO_3^-$. EC (OR 1.7 vs. 1.4) and OC (OR 1.6 vs. 1.4) also showed large differences between LBW and normal birth weight (Fig. 3). Exactly similar patterns were observed for $PM_{2.5}$ emission attributable to different sectors when stratified by LBW (Fig. 4).

### Potential nonlinear association

The exposure-response relationship of components OC, $NO_3^-$, and $NH_4^+$ exhibited a monotonic increase up to a midrange concentration value, whereas the rest of the components, such as EC, soil, and others, showed an inverted U-shaped pattern for LBW (Supplementary Fig. 2). However, for anaemia (Supplementary Fig. 3), the majority of components, such as EC, $NH_4^+$, $NO_3^-$, OC, $SO_4^{2-}$, and others, showed a monotonic increase in the probability of anaemia prevalence with an increase in pollutant exposure. In contrast, for soil, the trend line

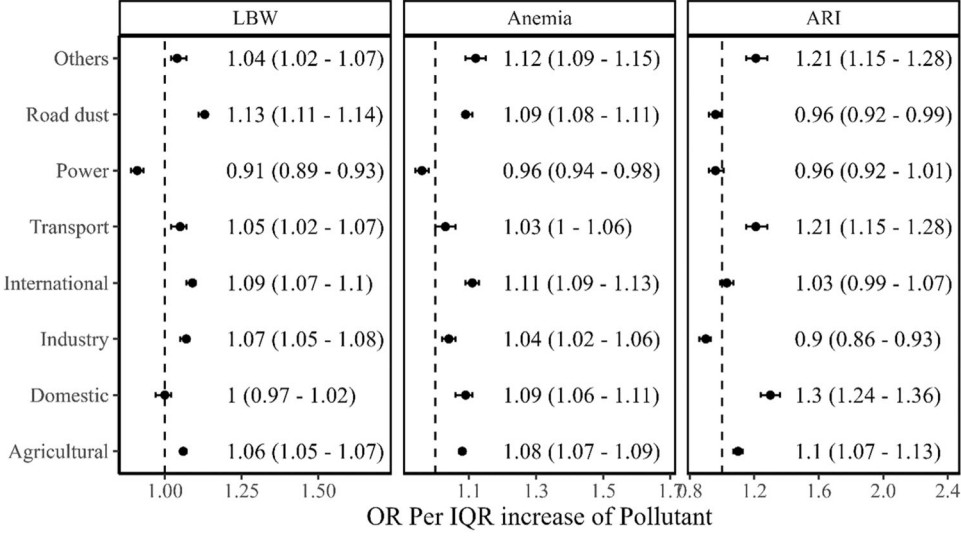

**Fig. 2 | Association between exposure to $PM_{2.5}$ contributed from various sectors and child health outcomes.** Odds ratio per IQR of $PM_{2.5}$ sectoral concentrations with their 95% confidence intervals (black dots with error bars) for given health outcomes i.e., LBW, Anaemia, and ARI. The analytical sample size used for each health outcome is provided in the consort diagram (Supplementary Fig. 1).

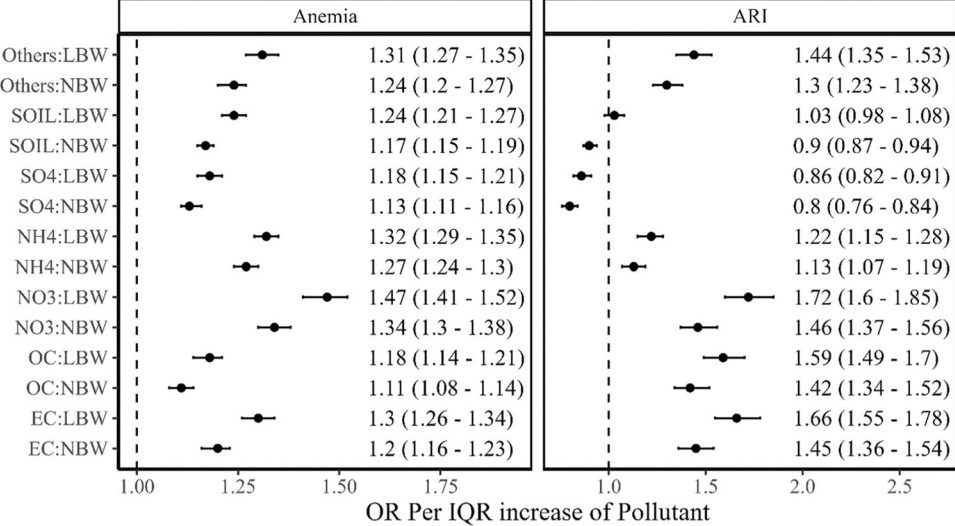

**Fig. 3 | Association between per IQR of $PM_{2.5}$ components concentrations with an interaction of birth weight status and health outcomes such as anaemia and ARI.** The odds ratios with their 95% confidence intervals are represented in black dots with error bars. LBW represents low birth weight; NBW represents normal birth weight. The analytical sample size used for each health outcome is provided in the consort diagram (Supplementary Fig. 1).

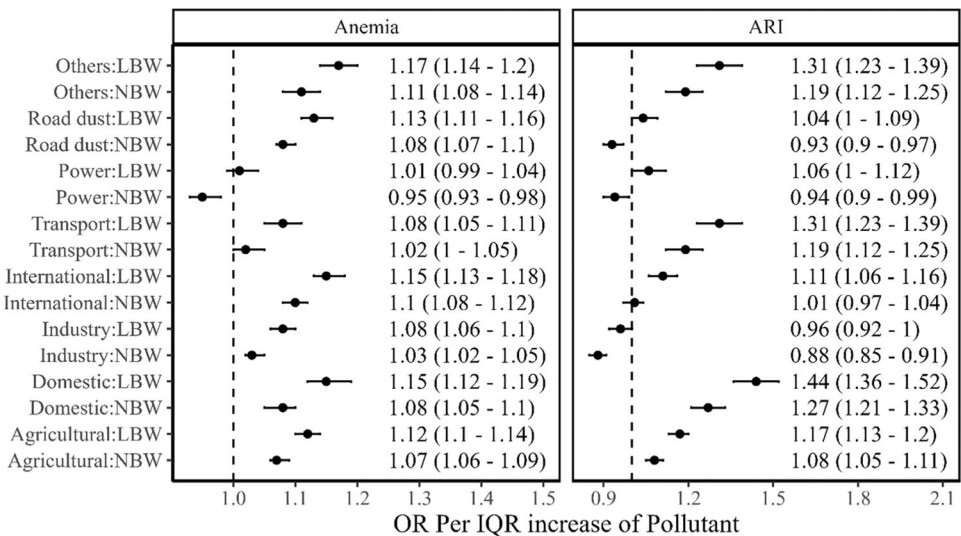

**Fig. 4 | Association between per IQR of sectoral PM_{2.5} concentrations with an interaction of birth weight status and health outcomes such as anaemia and ARI.** Odds ratios with their 95% confidence intervals are represented in black dots with error bars. LBW represents low birth weight; NBW represents normal birth weight. The analytical sample size used for each health outcome is provided in the consort diagram (Supplementary Fig. 1).

witnessed a fall. The nonlinear association between the probability of anaemia prevalence increased with increasing PM_{2.5} exposure. Similarly, varying nonlinear patterns were observed for ARI (Supplementary Fig. 4).

### Cumulative effects

The cumulative effects of PM_{2.5} components on respective child health endpoints were estimated by summing over individual component-specific estimates of regression coefficients ('Methods') adjusted to their respective mass fraction. The cumulative regression coefficient was defined as

$$\hat{\beta}_{cum} = \sum_{i=1}^{k} m_i \hat{\beta}_i$$

where, $m_i$ was the mass fractions and $\hat{\beta}_i$ was the estimated regression coefficient for $i$th component. The standard error of $\hat{\beta}_{cum}$ was estimated by $V(\hat{\beta}_{cum}) = \sum_{i=1}^{k} m_i^2 V(\hat{\beta}_i)$ assuming independence among regression coefficients of individual components.

The cumulative OR of PM_{2.5} components for $10\,\mu g\,m^{-3}$ increase in PM_{2.5} mass was estimated as 1.23 (1.21–1.26) for LBW, 1.49 (1.45–1.52) for anaemia and 1.35 (1.29–1.41) for ARI, while for every $10\,\mu g\,m^{-3}$ increase in PM_{2.5} mass, ORs were estimated as 1.05 (1.04–1.06), 1.10 (1.09–1.11), and 1.11 (1.08–1.13) for LBW, anaemia, and ARI, respectively.

### Expected health benefits of meeting clean air targets

The national ambient air quality standard (NAAQS) for annual PM_{2.5} was set at $40\,\mu g\,m^{-3}$ in 2009 by the Government of India[21]. We examined the expected health benefits of meeting the NAAQS (Fig. 5b, e, h) and eventually the WHO-AQG (Fig. 5c, f, i) based on the indigenous exposure-response functions. We first estimated the expected reduction in ambient PM_{2.5} exposure ($\triangle PM_{2.5}$) at the district level if India successfully meets the NAAQS and then the WHO-AQG relative to the current exposure level. Then, we calculated the attributable fraction (AF) as follows:

$$AF = \frac{(RR - 1)}{RR}$$

where $RR = \exp(\log(OR) \times \triangle PM_{2.5})$, and OR is the estimated odds ratio for each unit increase of PM_{2.5} exposure reported in previous

subsection (ORs were converted for unit increase of PM_{2.5}). Finally, we calculated the expected reduction in district-level prevalence (E) for LBW, anaemia, and ARI due to the reduction in PM_{2.5} exposure as:

$$E = AF \times \text{District level prevalence}$$

While we consider OR for elevation of PM_{2.5} mass (i.e., 1.005, 1.01 and 1.011 for LBW, anaemia and ARI, respectively) the overall LBW prevalence could reduce from 16.6% (16.4, 16.7) to 14.5% (14.1, 14.9) if the NAAQS level is achieved. If the exposure is reduced to the WHO-AQG level, the LBW prevalence could reduce to 11.6% (11.1, 12.4). For anaemia, the prevalence could reduce from 56.8% (56.6, 57.1) to 44.8% (43.8, 45.7) and further to 32.9% (32.1, 33.6), respectively, while for ARI, the prevalence could reduce from 2.8% (2.7, 2.9) to 2.1% (1.9, 2.3) and further to 1.5% (1.3, 1.6) if the NAAQS and WHO-AQG levels are achieved.

While we consider cumulative OR for elevation of PM_{2.5} components (i.e., 1.021, 1.041 and 1.03 for LBW, anaemia and ARI, respectively, the overall LBW prevalence could reduce from 16.6% (16.4, 16.7) to 15.7% (15.3, 16.1) if the NAAQS level is achieved. If the exposure is reduced to the WHO-AQG level, the LBW prevalence could reduce to 14.7% (14.3, 15.09). For anaemia, the prevalence could reduce from 56.8% (56.6, 57.1) to 50.7% (49.7, 51.8) and further to 44.2% (43.3, 45.1), respectively, while for ARI, the prevalence could reduce from 2.8% (2.7, 2.9) to 2.3% (2.1, 2.7) and further to 2.1% (1.8, 2.5) if the NAAQS and WHO-AQG levels are achieved.

## Discussion

In this study, we present, to our knowledge, the first comprehensive assessment of the effects of PM_{2.5} and its components, as well as contributing sectors, on three specific U5 children health outcomes in India. We chose LBW, anaemia, and ARI to represent the health burden of U5 children because these are common public health morbidities reported in India, have pathophysiological frameworks that include air pollution, and occur in disparate frameworks of time or organ systems. PM_{2.5} is a mixture made up of various components with different sources and toxicities. Each source may produce different PM_{2.5} components (either in primary form or secondary precursors), and each component of PM_{2.5} may likewise come from various sources. Our findings suggest that the major health risk was posed by exposure

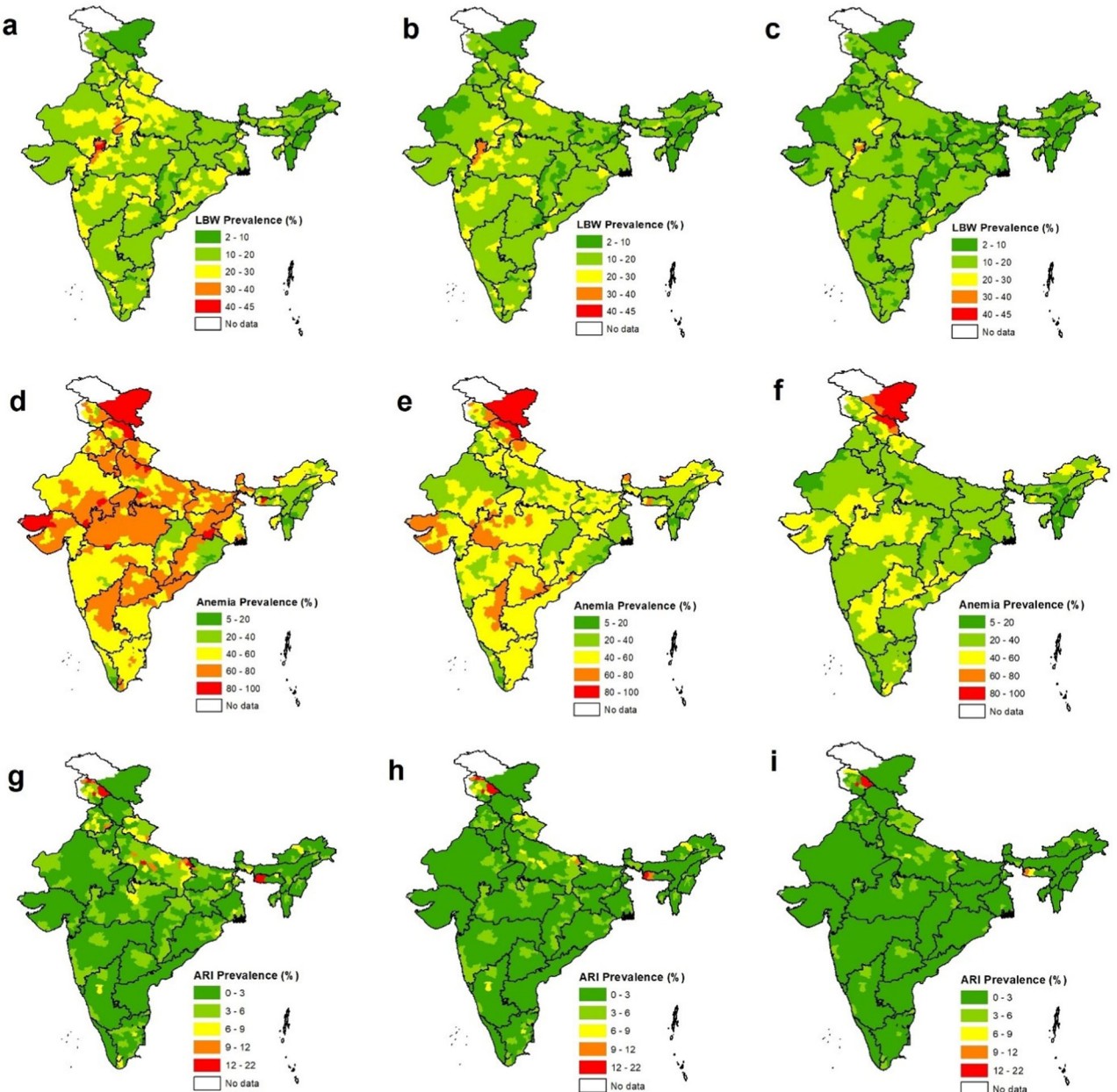

**Fig. 5 | Benefits of meeting clean air targets. a** Low birth weight prevalence (%) across Indian districts, **b** low birth weight prevalence (%) after NAAQS implementation, **c** low Birth weight prevalence (%) after next air quality standards implementation; **d** anaemia prevalence (%) across Indian districts, **e** anaemia prevalence (%) after NAAQS implementation, **f** anaemia prevalence (%) after next air quality standards implementation; **g** ARI prevalence (%) across Indian districts, **h** ARI prevalence (%) after NAAQS implementation, **i** ARI prevalence (%) after next air quality standards implementation.

to NO$_3^-$, NH$_4^+$, EC, and OC. However, other components also indicated a consistent risk to child health. Furthermore, we analysed the impact of sectoral PM$_{2.5}$ on all three child health outcomes. The PM$_{2.5}$ exposure attributable to the sectoral emissions from road dust, transport, industry, agriculture, domestic, and others, as well as international sources, depicted adverse effects on at least one health endpoint (see Fig. 2). The estimated varying impacts of different PM$_{2.5}$ components, as well as those attributable to sectoral emission, can be considered as the direct effects of each component on health, as when estimating the effect for one component, the covariation of the other components with health was adjusted indirectly by the combination of Stage-1 and Stage-2 models.

The estimate of the effects of PM$_{2.5}$ on LBW was quite similar to the estimates observed in the current literature[22,23]. Sun et al.[23]

observed significant effects of specific PM$_{2.5}$ components, including potassium, zinc, nickel, titanium, elemental carbon, silicon, and ammonium, to be more harmful than aggregated PM$_{2.5}$. Some studies have reported a significant positive association between anaemia and PM$_{2.5}$ among U5 children, but not much has been reported with PM$_{2.5}$ constituents[16,17]. Several studies have reported adverse effects of PM$_{2.5}$ on either occurrence or frequency of ARI in U5 children, but none have assessed the impact of different PM components to understand their true cumulative impact[24,25]. The true impact of every 10 μg m$^{-3}$ increase in PM$_{2.5}$, which was an additive effect of PM$_{2.5}$ chemical components (cumulative effect), was observed to be considerably high (ORs: 1.23 for LBW, 1.49 for anaemia, and 1.35 for ARI) as compared to the estimated effects by total PM$_{2.5}$ mass (ORs: 1.05 for LBW, 1.10 for anaemia, and 1.11 for ARI). Therefore, the total PM$_{2.5}$ mass, considered as the key

surrogate marker for air pollution exposure, could substantially underestimate the true composite effect of different components of $PM_{2.5}$.

There is a sufficiently large evidence base that the association of $PM_{2.5}$ and its components with LBW is possibly causal. It is known that $PM_{2.5}$ components can translocate into and cross the placental barrier and can induce oxidative stress that can cause placental changes[26]. Oxidative stress and placental inflammation may impair transplacental nutrient and oxygen exchange in the placenta, preventing enough nutrients and oxygen from reaching the foetus at the right time during gestation and impacting foetal growth[27,28]. Similarly, in U5 children, the continuous exposure of the lung to pro-oxidants through $PM_{2.5}$ or its components causes oxidative stress leading to prolonged inflammation and lowered immunity, which is a common physiological pathway for vulnerability to bacterial and viral infections[29,30]. The causality of anaemia through $PM_{2.5}$ or its components is probably linked to chronic inflammation. The secretion of inflammatory cytokines like interleukin-6 will signal the liver to secrete hepcidin, which reduces the absorption of dietary iron[31]. It does so by binding to ferroprotein, the cellular transmembrane iron-exporter, which prevents the internalization of dietary iron from the intestinal epithelium, as well as the recycling of iron within the body by macrophages. These events together reduce the amount of iron available for Hb synthesis. Equally, erythropoiesis can be independently reduced by the inflammatory cytokines[32].

Our results contribute to the sparse body of evidence about the impact of $PM_{2.5}$ components on health, particularly in LMICs, and show that the risk assessment purely based on $PM_{2.5}$ mass may significantly understate the impact of some of its more dangerous components. These findings call for additional laboratory research to completely comprehend the biological mechanisms through which the diverse $PM_{2.5}$ components affect human health. There are several limitations of this study. First, we assumed that the child's residence did not change during the early life exposure period. Second, we considered the mass fractions of each species and sectoral $PM_{2.5}$ from the model to be representative of the entire exposure duration. We combined this model data with satellite-derived $PM_{2.5}$ (see 'Methods') to get the estimates of the required duration covering gestational and early life exposures. The exposure-response curves were observed to be mostly nonlinear and non-monotonic, unadjusted confounding could be one of the causes. The true causes of the nonlinearity could be difficult to explain from this study based on a cross sectional national survey with a cluster level air pollution exposure. Therefore, the magnitude of the varying effects may require further validation with well-planned cohorts.

An additional aspect of our study is that it provides information on the benefits of meeting clean air targets on child health. The analysis implies that with continued efforts towards mitigating ambient air pollution, the health burden among the children population can be reduced. In fact, meeting the NAAQS would take the anaemia burden closer to the 'anaemia-free India' mission target (reducing the prevalence to 40%) of the Government. The sectoral analysis that we presented will be useful for policymakers. For example, the residential and industrial sectors were the major contributors to ambient $PM_{2.5}$, with shares of 41% and 37%, respectively, followed by agricultural residue burning (8%), other sectors (7%), transport (3%), and power (2.6%) (Supplementary Fig. S5)[33]. Accelerating the supply of clean energy for household activities through the Pradhan Mantri Ujjwala Yojana scheme, imposing stricter emission norms and gradual transition to clean energy usage in the industries, efficiently managing open burning are expected to provide a greater health benefit to the children. We note that the modelled mass fractions of individual species may have some uncertainties depending on the representativeness of emission inventory, which was also highlighted in the literature[33]. Nonetheless, this will not alter the broad conclusion of the study that

the cumulative impact of $PM_{2.5}$ components is greater than the impact of $PM_{2.5}$ mass on child health in India.

To summarize, in a first-of-its-kind study, we demonstrated a significant association between components and sectors contributing to $PM_{2.5}$ with LBW, anaemia, and ARI among U5 children in India. Our study further showed that the children born with LBW are more highly impacted due to $PM_{2.5}$ components exposure than the children born with normal birth weight. We recommend further epidemiological and toxicological studies to understand the biological pathways that drive the linkage between air pollution and its causal effects and use these data in driving clean air actions.

## Methods
### Health data
The health dataset was obtained from Demographic Health Survey (DHS) version seven https://www.dhsprogram.com/data/available-datasets.cfm, which provides national-level health data for India as the NFHS-4[34]. NFHS-4 was a household survey conducted between 20 January 2015 and 4 December 2016 across all 640 districts of India. The survey data provided the necessary information on health and family welfare, along with details on current threats in these areas, to aid policies and programmes in India's health sector over time. Information on socioeconomic status, reproductive health and family planning, maternal and child health, breastfeeding and nutrition, vaccination coverage, anaemia, and the symptoms of ARI were collected in the survey.

The NFHS-4 sample was a two-stage stratified sample. The sampling frame for the selection of Primary Sampling Units (PSU) was the 2011 census. PSUs in rural areas were villages, and Census Enumeration Blocks (CEBs) were in urban areas. PSUs with less than 40 households were combined with the closest PSU. Probability Proportional to Size (PPS) sampling was used to select the final PSUs. Selected PSUs with more than 300 households were divided into segments of 100–150 households, and two segments were selected at random with probability proportional to segment size. In the second stage, 22 households were selected from each rural and urban cluster using systematic sampling. Data collection was carried out using various questionnaires. The survey included four questionnaires—a household questionnaire, a woman's questionnaire, a man's questionnaire, and a biomarker questionnaire. More detailed information on the sample design and selection is available on the International Institute of Population Sciences *National Family and Health Survey-4 Report*[34].

### Ambient $PM_{2.5}$ exposure data
Our primary exposure metric was ambient $PM_{2.5}$ exposure. Monthly average $PM_{2.5}$ concentrations at PSU level were used to construct long-term exposure for three different health outcomes. For low birth weight, the average of monthly $PM_{2.5}$ ambient concentrations at the PSU during the pregnancy with number of months being derived based on individual gestational age was considered as pregnancy period exposure. For anaemia and ARI, the $PM_{2.5}$ exposure was derived by averaging monthly ambient $PM_{2.5}$ concentration at the PSU over the period of individual life course. We therefore, estimated the exposure throughout the gestational period and life course of U5 children across all 636 districts from the national $PM_{2.5}$ database created at a $1\,km \times 1\,km$ spatial scale for India[35]. This database was created by converting MODIS-MAIAC aerosol optical depth to surface $PM_{2.5}$ using a dynamic scaling factor from MERRA-2 reanalysis data. The instantaneous $PM_{2.5}$ (representing the satellite overpass time) was then converted to a 24-h average using the diurnal scaling factor from MERRA-2. Both these scaling factors were calibrated against the data from the existing ground-based network of the Central Pollution Control Board of India (CPCB). At the annual scale, satellite-derived $PM_{2.5}$ concentration showed a correlation coefficient of 0.97, and a root mean square error of $7.2\,\mu g\,m^{-3}$ with the coincident ground-based measurements from

the CPCB network. The Individual level exposure to ambient $PM_{2.5}$ was calculated by clustering the exposure using geocode information of each PSU in the NFHS-4 data.

## $PM_{2.5}$ composition
Satellite data cannot provide direct information on $PM_{2.5}$ composition, and very sparse ground measurements are available in India, that too for a very limited duration. Therefore, we integrated satellite-$PM_{2.5}$ data with outputs from a published study which employed WRF version 3.9.1-CMAQ version 5.3.1 setup to simulate ambient $PM_{2.5}$ concentrations at $36 \times 36$ km$^2$ spatial resolution with 25 vertical levels[33]. The modelling set-up (Supplementary Table 3) employed ERA5 meteorology and emissions estimated using Greenhouse Gas and Air Pollution Interactions and Synergies (GAINS)-ASIA model (https://gains.iiasa.ac.at) in which government reported energy consumption data for the different sectors was used as an input for the year 2016. ECLIPSE (version 5) database of IIASA (2014) had been used for the national ammonia emissions, ship emissions, and the emissions emitted from the neighbouring countries which fall within the study domain that, includes Bhutan, Nepal, Myanmar, Bangladesh, Sri Lanka, Pakistan, and parts of China and Afghanistan. In addition, transboundary pollutants coming from outside the study domain were taken from boundary conditions developed by the Community Atmosphere Model with Chemistry (CAM-chem) model (https://www.acom.ucar.edu/cam-chem/cam-chem.shtml). The ambient $PM_{2.5}$ simulated concentrations showed a significant agreement for the coefficient of determination when compared with the observed concentrations at ground-based monitoring stations[33]. The coefficient of determination between the observed versus simulated monthly averaged concentrations was found to be 0.81, while the index of agreement was 0.94.

We estimated the mass concentrations of each $PM_{2.5}$ component ($M_i$) as

$$M_i = \frac{M_{i,\text{model}}}{PM_{2.5,\text{model}}} \times PM_{2.5,\text{satellite}} \qquad (1)$$

where $M_{i,\text{model}}$ and $PM_{2.5,\text{model}}$ are the mass of component '$i$' and total $PM_{2.5}$ mass derived from the model and $PM_{2.5,\text{satellite}}$ is the satellite-derived $PM_{2.5}$. Since the model estimates were available at 36 km $\times$ 36 km resolution, we bilinearly re-gridded the model data to 1 km $\times$ 1 km to match its resolution with the satellite-based $PM_{2.5}$ dataset resolution. Once the gridded mass concentrations of each component were estimated, we calculated the exposure for the NFHS clusters.

## Sectoral contribution to $PM_{2.5}$
The sectoral contributions to annual $PM_{2.5}$ were estimated by the subtraction method in the modelling framework. First, the control simulation was carried out with all sectoral emissions on, and then in each subsequent simulation, emissions from a particular sector were switched off, and the difference provided the contribution from that sector. The model outputs[33] were analysed to derive $PM_{2.5}$ contributions of the transport, small and medium-scale industries, brick industry, major industries, power, domestic (due to solid fuel used for cooking and heating), agriculture residue burning, construction, road dust, and others (which includes refuse burning, construction, crematoria, $NH_3$, biogenic emissions, refineries, and evaporative non-methane volatile organic compounds). Here we used these model outputs and combined them with satellite-$PM_{2.5}$ following the method explained in the previous section (for $PM_{2.5}$ components) to estimate the exposure to a specific sectoral $PM_{2.5}$ for the NFHS clusters.

## Outcome data
For U5 children, the primary health outcomes considered in this study were LBW, anaemia, and ARI. For LBW, the NFHS-4 recorded birth weight from either written records or the mother's oral report. LBW was defined by the WHO as a birth weight <2500 g[36]. Children with LBW were coded as 1, whereas the children with birth weight greater than 2500 g were coded as 0.

Information on anaemia prevalence among U5 children was obtained from NFHS-4, which used the finger or heel prick method to collect blood samples. Haemoglobin concentrations were measured on-site using the HemoCue Hb 201+ analyzer[37]. Children with haemoglobin levels <11 g/dL were considered anaemic, and children with haemoglobin levels >11 g/dL were considered non-anaemic[38]. These were coded as 1 and 0, respectively.

ARI in U5 children was diagnosed by the reporting of symptoms like cough, accompanied by short rapid breathing and/or difficulty in breathing that was thought to be chest related. The interviewer asked mothers whether their children experienced any ARI symptoms in the two weeks preceding the survey. ARI was used as a dichotomous variable with the presence of ARI symptoms coded as 1 and the absence of ARI symptoms coded as 0.

## Covariates
Several individual-level and household-level variables were identified as potential covariates of LBW[39,40], anaemia[41] and ARI[42]. These variables were included in the analysis based on their significance with the respective health outcome. The following individual-level variables were considered for all three outcome variables: sex of the child (male or female), mother's education (no education, primary, secondary, and higher), and parity (1, or >1). In addition, for the outcome variable ARI, we accounted for the age of the child, for LBW, mother's age (<20, 20–35, >35) and body mass index (BMI) of the mother (underweight, normal weight, overweight and obesity), lastly, for anaemia, maternal haemoglobin levels, and per capita iron intake. Daily dietary iron intake (per capita) was obtained by converting monthly food purchases captured by the 9th quinquennial Household Consumer Expenditure survey of the 68th round of the NSSO[43] which was further triangulated with NFHS-4 by Swaminathan et al.[44]. Household-level covariates used for both ARI and anaemia were the following: socioeconomic status, which is classified into five wealth quintiles (poorest, poor, middle, rich, and richest), type of residence (rural or urban), and passive smoking (yes or no). Since studies have shown that ambient[20] and household[36] air pollution are associated with anaemia, we also included the type of cooking fuel in the household as a covariate. This was classified as clean fuel (electricity, LPG/natural gas, biogas), solid fuel (coal/lignite, charcoal, wood, straw/shrubs/grass, agricultural crop waste, and dung cakes), kerosene, and others. For LBW, only the wealth index was used as the household-level covariate.

## Statistical analysis
Gestational period exposure was associated with LBW, and life course exposure was associated with ARI and anaemia. PSU level ambient concentration was assigned as the exposure to air pollutants for children residing within the PSU. To account for this cluster effect, we used a logistic mixed effects regression model. We also wanted to estimate the direct effects of all $PM_{2.5}$ components adjusted for other covariates. Owing to high collinearity among the $PM_{2.5}$ components, we used a two-stage model to avoid potential multi-collinearity. In the first step, we regressed $PM_{2.5}$ mass concentration on each component at a time and extracted the residuals as an alternative metric that can capture the total variability of $PM_{2.5}$ except variability explained by the component. These $PM_{2.5}$-residuals were then adjusted in the second stage logistic mixed model.

Stage-I:

$$(\text{ResPM}_{2.5})_i = PM_{2.5} - \hat{\delta}_0 - \hat{\delta}_1 P_i$$

Stage-II:

$$\text{logit}\left\{\text{Prob}\left(Y_{ij}=1|u_i\right)\right\}=\beta_0+\beta_1 P_i+\beta_2\left(\text{ResPM}_{2.5}\right)_i+\gamma(\text{confounders})_{ij}+u_i$$

Where $\hat{\delta}_0$ and $\hat{\delta}_1$ were the estimates of intercept and slope from the first stage model. $Y_{ij}$ represented the binary outcome (anaemia/LBW/ARI) for the $j$th individual in $i$th PSU, $P_i$ was one of the PM$_{2.5}$ components, $(\text{ResPM}_{2.5})_i$ was the residuals at $i$th PSU, $u_i$ was a random intercept corresponding to $i$th PSU with $u_i \sim N(0,\sigma_u^2)$ and $\gamma(\text{confounders})_{ij}$ represents the linear terms for all confounders adjusted.

Further, the estimates of PM$_{2.5}$ components on anaemia and ARI were stratified by birth weight status (LBW vs. Normal) using an interaction component of LBW with the component in 2nd stage model. Stratified analysis was performed by adding an interaction term to the stage-II model. Potential nonlinear associations of pollutants on health outcomes were explored by replacing $P_i$ with a penalized cubic smoothing spline function of $P_i$ [i.e., $f(P_i)$] in the stage-II model.

To compare PM$_{2.5}$ components specific estimate against the effects estimate for elevation of whole PM$_{2.5}$ mass we also estimated adjusted OR of PM$_{2.5}$ mass for LBW, anaemia and ARI by cluster-logistic regression, cluster being the PSU.

R version 4.1.2 (R Core Team, 2022 Vienna, Austria)[45] was used for all statistical analyses.

### Reporting summary
Further information on research design is available in the Nature Portfolio Reporting Summary linked to this article.

## Data availability
The Demographic Health Survey data used in this study can be accessed through a restricted access system in accordance with the data access guidelines of the DHS programme. To obtain access, a request should be submitted after registering on the DHS website at the following link: https://dhsprogram.com/data/Using-Datasets-for-Analysis.cfm. Source data are provided as a Source data file and have also been deposited in figshare under accession code https://doi.org/10.6084/m9.figshare.23513757[46]. Exposure datasets used in this study can be accessed using the same figshare accession code. Source data are provided with this paper.

## Code availability
R codes for the statistical models in the main text are available at https://doi.org/10.6084/m9.figshare.23513757. R codes for data processing techniques are available on request from the corresponding authors.

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

## Acknowledgements

The work is supported by a research grant under the SUPRA scheme from the SERB, Department of Science and Technology, GoI (SPR/2020/000212). The exposure database was generated as part of a project under the NCAP funded by the Ministry of Environment, Forest and Climate Change, GoI. S.D. acknowledges financial support for the Institute Chair fellowship and DST-FIST programme (SR/FST/ESII-016/2014) for computing support.

## Author contributions

E.C., F.G., S.D. and S.G. conceived the study and wrote the initial draft of the article. E.C., F.G. and A.S. analysed the data with the help of S.G. and S.D. S.S., N.S. and S.A. did the model simulations and analysis of the model outputs. A.V.K. and T.T. helped to review the potential causal pathways of the observed association. U.M. provided essential advice about the anaemia outcome. All authors provided comments and contributed to give the final shape of the article.

## Competing interests

The authors declare no competing interests.
