## [Peer review file · Nature Communications]

REVIEWER COMMENTS

Reviewer #1 (Remarks to the Author):

This paper describes a large cohort epidemiological quantification of the effects of PM_{2.5} components and their contributing sources on three neonatal/young child adverse health outcomes in India. The three outcomes are low birth weight, and anaemia and acute respiratory infection in under 5s. Individual (child and mother) and household-level data on health and potential modifiers and confounders are available for 152,000 to 177,000 instances (numbers differ according to health outcome). The exposures to total PM_{2.5} and to chemical components within the PM_{2.5} are estimated from a combination of satellite data and WRF-CMAQ chemical transport modelling. The PM_{2.5} contributions by source are estimated by modelling (using the 'brute force' approach of switching off sources in turn).

The paper deals with a topic of very substantial societal impact and policy implication. Exposure to PM_{2.5} is well known to cause the greatest mortality and morbidity of any environmental hazard globally. 'Start of life' is part of the life course with likely greater susceptibility to PM_{2.5} pollution than other parts. The extent of differential toxicity of different PM_{2.5} components and sources remains unresolved. The population of India is generally exposed to much greater PM_{2.5} concentrations than those in developed countries so there is clear need to quantify the burdens and to identify where mitigation is best focused. This study makes a substantial contribution to the knowledge base for these child health impacts in India.

Description of methods and results is well written and focused. The methods used appear appropriate. There are always limitations in estimating PM_{2.5} total and PM_{2.5} compositional exposure for these sorts of studies. The authors describe some of these limitations. Different methodological approaches will yield different estimates of PM_{2.5} exposure, and different splits of the chemical and source contributions to the exposure, but such differences are unlikely to alter the broad scientific findings.

The following are two areas where the authors should include additional explanatory discussion.

- (1) The substantial curvatures in the OR plots shown in Supplementary Figures S2-S4.
- (2) The realism or not of the very small contributions of elemental carbon (EC) to all sources, particularly to those where EC emissions are expected such as transport, power, and waste burning.

L509: the word "interaction" is duplicated.

Reviewer #2 (Remarks to the Author):

This manuscript analyzed the health effect (acute respiratory infection and low birth rate) associated with PM exposure on children in India. The topic of health effect of PM_{2.5} is relevant to the scope of Nature Communications, but there are some major issues in the draft and the draft is hard for readers to follow. I would only recommend this manuscript to be accepted with major modification.

Major concerns

Lines 294 to 315, the description of model application is limited. Readers cannot find detail settings of the model like domain settings, resolution, performance evaluation etc. This made the simulation process like a black box.

Lines 165 to 167, I could not find how OR is calculated at Method section and Lines 343 to 358 is too general without any detailed information for how health data is used.

Reviewer #3 (Remarks to the Author):

I congratulate the authors in conducting this secondary analysis of data on exploring the estimation of impact of PM2.5 and its constituent components.

I have few observations in your manuscript (not in any order of priority or importance), as follows:

1. WHO estimates that 99% of the population are living in areas exposed to higher than 5Mcg/m³ of PM2.5. This is the current understanding. Please correct as you have mentioned 90%.
2. Editorial comment. Need to improve the flow in write up such as the following should come early in the para and then focusing on children's health impact. "The latest round of the state-level burden of diseases in India estimated that ambient PM2.5 exposure was responsible for 0.98 million (0.77-1.19) deaths and 17.8 million disability-adjusted life years."
3. Why the transport sector has the lowest impact on LBW and higher for ARI? Are these random variation in data or some specific reason? These variations occur in the estimates need to have good discussion with references, if possible. "Among the eight PM2.5 sources analyzed using the two-stage model, we found that the IQR increase in PM2.5 from road dust exhibited a higher effect on LBW (OR: 1.13, 1.11- 1.14), followed by the international transboundary transport (OR: 1.09, 1.07 - 1.1), the industry sector (OR: 1.07, 1.05 - 1.08), the agricultural sector (OR: 1.06, 1.05- 1.07), other sectors (OR: 1.04, 1.02 - 1.07) and transport sector (OR: 1.05, 1.02 -114 1.07)."
4. Exposure were average over pregnancy length (what about variable duration of pregnancy length) and average over child age since birth [Was it monthly data or daily data?]
5. Now new DHS data is available as you are utilizing decade old data for estimating the health impact which may not be relevant in 2023.

I would appreciate addressing or responding to these comments.

Reviewer's comments

Responses to the reviewers' comments are provided below in blue colour fonts.

Reviewer #1

This paper describes a large cohort epidemiological quantification of the effects of PM2.5 components and their contributing sources on three neonatal/young child adverse health outcomes in India. The three outcomes are low birth weight, and anaemia and acute respiratory infection in under 5s. Individual (child and mother) and household-level data on health and potential modifiers and confounders are available for 152,000 to 177,000 instances (numbers differ according to health outcome). The exposures to total PM2.5 and to chemical components within the PM2.5 are estimated from a combination of satellite data and WRF-CMAQ chemical transport modelling. The PM2.5 contributions by source are estimated by modelling (using the 'brute force' approach of switching off sources in turn). The paper deals with a topic of very substantial societal impact and policy implication. Exposure to PM2.5 is well known to cause the greatest mortality and morbidity of any environmental hazard globally. 'Start of life' is part of the life course with likely greater susceptibility to PM2.5 pollution than other parts. The extent of differential toxicity of different PM2.5 components and sources remains unresolved. The population of India is generally exposed to much greater PM2.5 concentrations than those in developed countries so there is clear need to quantify the burdens and to identify where mitigation is best focused. This study makes a substantial contribution to the knowledge base for these child health impacts in India.

Description of methods and results is well written and focused. The methods used appear appropriate. There are always limitations in estimating PM2.5 total and PM2.5 compositional exposure for these sorts of studies. The authors describe some of these limitations. Different methodological approaches will yield different estimates of PM2.5 exposure, and different splits of the chemical and source contributions to the exposure, but such differences are unlikely to alter the broad scientific findings.

We thank the reviewer for appreciating our work and providing insightful comments. We have addressed them. The detailed responses are mentioned below.

The following are two areas where the authors should include additional explanatory discussion.

1) The substantial curvatures in the OR plots shown in Supplementary Figures S2-S4.

The exposure-response relationship of components OC, NO₃⁻, and NH₄⁺ exhibited a monotonic increase up to a midrange concentration value, whereas the rest of the components, such as EC, soil, and others, showed an inverted U-shaped pattern for LBW (Supplementary Fig. 2). However, for anaemia (Supplementary Fig. 3), the majority of components, such as EC, NH₄⁺, NO₃⁻, OC, SO₄²⁻, and others, showed a monotonic increase in the probability of anaemia prevalence with an increase in pollutant exposure. In contrast, for soil, the trend line witnessed a fall. However, these nonparametric curves are very much data-driven, so it is challenging to provide a comprehensive explanation of the observed curvature in this study, particularly when estimating the exposure at the cluster level. Therefore, we emphasized on the effects estimate at a linear scale and reported this as an exploration. We have now added two lines, stating this limitation of the study (lines 239-242).

Lines 239-242: The exposure-response curves were observed to be mostly nonlinear and non-monotonic, unadjusted confounding could be one of the causes. The true causes of the nonlinearity could be difficult to explain from this study based on a cross-sectional national survey with a cluster level air pollution exposure.

(2) The realism or not of the very small contributions of elemental carbon (EC) to all sources, particularly to those where EC emissions are expected such as transport, power, and waste burning.

Response: We thank the reviewer for this comment. It helped us to re-check the estimates, and we figured out a minor plotting error in the previous figure. We have now updated the supplementary Fig. 5 (also attached below). Even in the revised plot, the EC contributions are relatively lower than some of the other components of PM_{2.5}. This is because the plots depict not just the primary emissions of PM_{2.5} but also include the contribution of secondary inorganic particulates which are formed due to the interaction of gases like NO_x, SO_x, and NH₄. The sectors like transport and power are one of the biggest emitters of gaseous pollutants, i.e., NO_x and SO_x, respectively. These gaseous pollutant result in the formation of secondary particulates in the form of nitrates and sulphates, causing a decrease in the relative contribution of primary particulates like EC in total ambient PM_{2.5} concentrations. Further, another reason for the estimated lower EC contribution is the underlying uncertainty in the bottom-up emission inventory fed into the model. Several studies (e.g., Singh et al., 2021, Venkataraman et al., 2018; Verma et al., 2017) have also reached similar conclusions that modelled BC fraction (in total PM_{2.5}) in India usually remains underestimated. We hope that improvement in the BC emission monitoring technique leading to refinement in future emission updates will fix this issue. Currently, this is the best data we have, and we mentioned this in the Discussion (lines 254-257). We feel that the broad conclusion will not be altered due to this.

Lines 254-257: We note that the modelled mass fractions of individual species may have some uncertainties depending on the representativeness of emission inventory, which was also highlighted in the literature.³³ Nonetheless, this will not alter the broad conclusion of the study that the cumulative impact of PM_{2.5} components is greater than the impact of PM_{2.5} mass on child health in India.

L509: the word "interaction" is duplicated.

Response: Corrected

Reviewer #2

This manuscript analyzed the health effect (acute respiratory infection and low birth rate) associated with PM exposure on children in India. The topic of health effect of PM_{2.5} is relevant to the scope of Nature Communications, but there are some major issues in the draft and the draft is hard for readers to follow. I would only recommend this manuscript to be accepted with major modification.

We thank the reviewer for providing insightful comments. We have addressed them. The detailed responses are mentioned below.

Major concerns

Lines 294 to 315, the description of model application is limited. Readers cannot find detail settings of the model like domain settings, resolution, performance evaluation etc. This made the simulation process like a black box.

Response:

The model speciation dataset used in the study is a published dataset taken from Singh et al., 2020. We realize and agree that some details would make the readers follow the discussion easily. We have now mentioned details about the domain setting, resolution, and performance evaluation in lines 303 to 313. Also, we have added a table (Supplementary Table 3; also attached below) in the supplementary to provide model setup configuration.

Lines 303 to 313 Satellite data cannot provide direct information on PM_{2.5} composition, and very sparse ground measurements are available in India, that too for a very limited duration. Therefore, we integrated satellite-PM_{2.5} data from a published study which employed WRF version 3.9.1-CMAQ version 5.3.1 modelling set up to simulate ambient PM_{2.5} concentrations at 36 x 36 km² spatial resolution with 25 vertical levels.³³ The modelling set-up (Supplementary Table 3) employed ERA5 meteorology and emissions estimated using Greenhouse Gas and Air Pollution Interactions and Synergies (GAINS)-ASIA model (<https://gains.iiasa.ac.at>) in which government reported energy consumption data for the different sectors was used as an input for the year 2016. ECLIPSE (version 5) database of IIASA (2014) had been used for the national ammonia emissions, ship emissions, and the emissions emitted from the neighbouring countries which fall within the study domain that, includes Bhutan, Nepal, Myanmar, Bangladesh, Sri Lanka, Pakistan, and parts of China and Afghanistan. Additionally, transboundary pollutants coming from outside the study domain were taken from boundary conditions developed by the Community Atmosphere Model with Chemistry (CAM-chem) model (<https://www.acom.ucar.edu/cam-chem/cam-chem.shtml>). The ambient PM_{2.5} simulated concentrations showed a significant agreement for the coefficient of determination when compared with the observed concentrations at ground-based monitoring stations.³³ The coefficient of determination between the observed versus simulated monthly averaged concentrations was found to be 0.81, while the index of agreement was 0.94.

Supplementary Table 3: WRF-CMAQ Model set-up details

Data/scheme type	Options
Model resolution	36 x 36 km with 25 vertical levels
Meteorology data used	ECMWF's ERA-5
Simulation period	The model was simulated from 1 st Jan 2016 to 31 st Dec 2016.
Micro Physics Options in WRF	WSM three-class simple ice scheme ¹
Shortwave radiation in WRF	Dudhia Shortwave Scheme ²
Longwave radiation in WRF	RRTM Longwave Scheme ³
Surface layer	Revised MM5 Monin-Obukhov scheme ⁴
Land Surface Options in WRF	Unified Noah Land Surface Model scheme ⁵
Boundary layer Options in WRF	ACM2 ⁶
Cumulus Options in WRF	Kain-Fritsch (new Eta) scheme ⁷
Chemical mechanism used in CMAQ	CB6r3_ae7_aq (CB6r3 - Carbon Bond 6 version r3; ae7_aq module - CMAQ's aero7 for treatment of SOA set up for standard cloud chemistry) ^{8,9}
Emission inventory resolution	36x36 km
Emissions of neighbouring countries within the study domain	ECLIPSE (version 5) database of IIASA
Transboundary pollutant outside the study domain	Community Atmosphere Model with Chemistry (CAM-chem) model

Lines 165 to 167, I could not find how OR is calculated at Method section and Lines 343 to 358 is too general without any detailed information for how health data is used.

Response: We thank the reviewer for raising this issue. We have now added a few lines (167,173,180; corresponding to 165-167) that must clarify the concern. These ORs were stated in the previous subsection (lines 147-154). The method is also included in the 'method' section now (lines 393-395)

Lines 167-186:

where $RR = \exp(\log(OR) \times \Delta PM_{2.5})$, and OR is the estimated odds ratio for each unit increase of $PM_{2.5}$ exposure reported in previous subsection (ORs were converted for unit increase of $PM_{2.5}$). Finally, we calculated the expected reduction in district-level prevalence (E) for LBW, anaemia, and ARI due to the reduction in $PM_{2.5}$ exposure as:

$$E = AF \times \text{District level prevalence}$$

While we consider OR for elevation of $PM_{2.5}$ mass (i.e., 1.005,1.01 and 1.011 for LBW, anaemia and ARI respectively) the overall LBW prevalence could reduce from 16.6% (16.4, 16.7) to 14.5% (14.1, 14.9) if the NAAQS level is achieved. If the exposure is reduced to the

WHO-AQG level, the LBW prevalence could reduce to 11.6% (11.1, 12.4). For anaemia, the prevalence could reduce from 56.8% (56.6, 57.1) to 44.8% (43.8, 45.7) and further to 32.9% (32.1, 33.6), respectively, while for ARI, the prevalence could reduce from 2.8% (2.7, 2.9) to 2.1% (1.9, 2.3) and further to 1.5% (1.3, 1.6) if the NAAQS and WHO-AQG levels are achieved.

While we consider cumulative OR for elevation of PM_{2.5} components (i.e., 1.021, 1.041 and 1.03 for LBW, anaemia and ARI respectively, the overall LBW prevalence could reduce from 16.6% (16.4, 16.7) to 15.7% (15.3, 16.1) if the NAAQS level is achieved. If the exposure is reduced to the WHO-AQG level, the LBW prevalence could reduce to 14.7% (14.3, 15.09). For anaemia, the prevalence could reduce from 56.8% (56.6, 57.1) to 50.7% (49.7, 51.8) and further to 44.2% (43.3, 45.1), respectively, while for ARI, the prevalence could reduce from 2.8% (2.7, 2.9) to 2.3% (2.1, 2.7) and further to 2.1% (1.8, 2.5) if the NAAQS and WHO-AQG levels are achieved.

Lines 393-395: To compare PM_{2.5} components specific estimate against the effects estimate for elevation of whole PM_{2.5} mass we also estimated adjusted OR of PM_{2.5} mass for LBW, anaemia and ARI by cluster-logistic regression, cluster being the PSU.

We have now modified the entire subsection of covariate selection (Lines 353 to 369 corresponding to lines 343-358) which has now detailed specification of health outcome-specific covariates.

Lines 353-369:

Covariates. Several individual-level and household-level variables were identified as potential covariates of LBW^{39,40}, anaemia⁴¹, and ARI⁴². These variables were included in the analysis based on their significance with the respective health outcome. The following individual-level variables were considered for all three outcome variables: sex of the child (male or female), mother's education (no education, primary, secondary, and higher), and parity (1, or >1). Additionally, for the outcome variable ARI, we accounted for the age of the child, for LBW, mother's age (<20, 20-35, >35) and body mass index (BMI) of the mother (underweight, normal weight, overweight and obesity), lastly, for anaemia, maternal haemoglobin levels, and per capita iron intake. Daily dietary iron intake (per capita) was obtained by converting monthly food purchases captured by the 9th quinquennial Household Consumer Expenditure survey of the 68th round of the NSSO⁴³ which was further triangulated with NFHS-4 by Swaminathan et al., 2019.⁴⁴ Household-level covariates used for both ARI and anaemia were the following: socioeconomic status, which is classified into five wealth quintiles (poorest, poor, middle, rich, and richest), type of residence (rural or urban), and passive smoking (yes or no). Since studies have shown that ambient²⁰ and household³⁶ air pollution are associated with anaemia, we also included the type of cooking fuel in the household as a covariate. This was classified as clean fuel (electricity, LPG/natural gas, biogas), solid fuel (coal/lignite, charcoal, wood, straw/shrubs/grass, agricultural crop waste, and dung cakes), kerosene, and others. For LBW, only the wealth index was used as the household-level covariate.

Reviewer #3

I congratulate the authors in conducting this secondary analysis of data on exploring the estimation of impact of PM_{2.5} and its constituent components.

I have few observations in your manuscript (not in any order of priority or importance), as follows:

We thank the reviewer for appreciating our work and providing insightful comments. We have addressed them. The detailed responses are mentioned below.

1. WHO estimates that 99% of the population are living in areas exposed to higher than 5Mcg/m³ of PM_{2.5}. This is the current understanding. Please correct as you have mentioned 90%.

Response: Thanks for pointing this out. It was a typo error. Now we have replaced 90% to 99% (line 41).

2. Editorial comment. Need to improve the flow in write up such as the following should come early in the para and then focusing on children's health impact. "The latest round of the state-level burden of diseases in India estimated that ambient PM_{2.5} exposure was responsible for 0.98 million (0.77-1.19) deaths and 17.8 million disability-adjusted life years."

Response: Agree with the suggestion. Required modifications are implemented (lines 47,48).

3. Why the transport sector has the lowest impact on LBW and higher for ARI? Are these random variation in data or some specific reason? These variations occur in the estimates need to have good discussion with references, if possible. "Among the eight PM_{2.5} sources analyzed using the two-stage model, we found that the IQR increase in PM_{2.5} from road dust exhibited a higher effect on LBW (OR: 1.13, 1.11- 1.14), followed by the international transboundary transport (OR: 1.09, 1.07 – 1.1), the industry sector (OR: 1.07, 1.05 - 1.08), the agricultural sector (OR: 1.06, 1.05– 1.07), other sectors (OR: 1.04, 1.02 – 1.07) and transport sector (OR: 1.05, 1.02 –1.07)."

Response: Proper explanation for this is difficult at this stage with a modelled data set. However, one possible reason could be the nature of the response of health to air pollution exposure. ARI is an acute response that could be higher for acute and highly variable PM_{2.5} exposure sourced from the transport sector. If the reviewer permits, we would like to refrain from stating such an explanation at this stage, rather we want the reader to interpret this pattern as an observation that needs to be confirmed by a prospective cohort study later.

4. Exposure were average over pregnancy length (what about variable duration of pregnancy length) and average over child age since birth [Was it monthly data or daily data?]

Response: We are sorry for the confusion. We have now modified the section such that the exposure assessment (lines 285-291) process gets clear to all. The average period was not

the same for everyone. It was decided based on individual gestational age only. For LBW, the exposure was averaged over the pregnancy period using pregnancy duration (in months) and month of delivery, while for the ARI and anaemia, the exposure was averaged for the early-life duration from the birth month to the month of the survey.

Lines 285-291:

Ambient PM_{2.5} exposure data. Our primary exposure metric was ambient PM_{2.5} exposure. Monthly average PM_{2.5} concentrations at PSU level were used to construct long term exposure for three different health outcomes. For low birth weight, the average of monthly PM_{2.5} ambient concentrations at the PSU during the pregnancy with number of months being derived based on individual gestational age was considered as pregnancy period exposure. For anemia and ARI, the PM_{2.5} exposure was derived by averaging monthly ambient PM_{2.5} concentration at the PSU over the period of individual life course. We therefore,

5. Now new DHS data is available as you are utilizing decade old data for estimating the health impact which may not be relevant in 2023.

Response: It is a great suggestion. However, due to exposure data (PM_{2.5} composition) constraints for NFHS-5, we considered NFHS-4 happened during 2015-2016, which temporally matches with CMAQ data (2016). The latest survey of DHS for India i.e., NFHS-5 happened during 2019-2021 and we have not yet developed the composition data for this period. Extrapolation of the same 2016 model data may end up with a biased estimation for NFHS-5. This can be a future study.

REVIEWERS' COMMENTS

Reviewer #1 (Remarks to the Author):

This is a revised paper for which I was one of the original reviewers.

I was supportive of the paper at its original submission (see original review). I asked the authors to respond to a couple of points, which they have done so adequately with inclusion of additional text.

I am happy to recommend publication.

Reviewer #2 (Remarks to the Author):

My comments were fully stressed now. Now I can recommend the draft to be accepted.

Reviewer's comments

Reviewer #1 (Remarks to the Author):

I was supportive of the paper at its original submission (see original review). I asked the authors to respond to a couple of points, which they have done so adequately with inclusion of additional text.

I am happy to recommend publication.

We thank the reviewer for appreciating and recommending our work for publication.

Reviewer #2 (Remarks to the Author):

My comments were fully stressed now. Now I can recommend the draft to be accepted.

We thank the reviewer for recommending our work for publication.